# SEER: Towards Efficient Preference-based Reinforcement Learning via Aligned Experience Estimation

## Abstract

One of the challenges in reinforcement learning lies in the meticulous design of a reward function that quantifies the quality of each decision as a scalar value. Preference-based reinforcement learning (PbRL) provides an alternative approach, avoiding reward engineering by learning rewards based on human preferences among various trajectories. PbRL involves sampling informative trajectories, learning rewards from preferences, optimizing policies with learned rewards, and subsequently generating higher-quality trajectories for the next iteration, thereby creating a virtuous circle. Distinct problems lie in effective reward learning and aligning the policy with human preferences, both of which are essential for achieving efficient learning. Motivated by these considerations, we propose an efficient preference-based RL method, dubbed SEER. We leverage state-action pairs that are well-supported in the current replay memory to bootstrap an empirical Q function ($\widehat{Q}$), which is aligned with human preference. The empirical Q function helps SEER to sample more informative pairs for effective querying and regularizes the neural Q function ($Q_\theta$), thus leading to a policy that is more consistent with human intent. Theoretically, we show that the empirical Q function is a lower bound of the oracle Q under human preference. Our experimental results over several tasks demonstrate that the empirical Q function is beneficial for preference-based RL to learn a more aligned Q function, outperforming state-of-the-art methods by a large margin.

## 1 Introduction

Deep Reinforcement Learning (DRL) has recently demonstrated remarkable proficiency in enabling agents to excel in complex behaviors across diverse domains, including robotic control and manipulation (Lillicrap et al., 2016; Gu et al., 2017), game playing (Mnih et al., 2013; Vinyals et al., 2019), and industrial applications (Xu & Yu, 2023). The foundation of success lies in providing a well-designed reward function. However, setting up a suitable reward function has been challenging for many reinforcement learning applications (Yahya et al., 2017; Schenck & Fox, 2017). The quality of the reward function depends heavily on the designer's understanding of the core logic behind the problem and relevant background knowledge. For example, formulating a reward function for text generation presents a significant challenge due to the inherent difficulty in quantifying text quality on a numerical scale (Wu et al., 2021b; Ouyang et al., 2022). Despite the substantial efforts of expert engineers in reward engineering, previous research (Victoria et al., 2020; Skalse et al., 2022) has highlighted various challenges, including phenomena like "reward hacking". In these scenarios, agents focus solely on maximizing their rewards by exploiting misspecification in the reward function, often leading to unintended and potentially problematic behaviors.

Recently, preference-based reinforcement learning (PbRL) has gained widespread attention and has given rise to a series of algorithms (Lee et al., 2021b; Park et al., 2022; Liu et al., 2022). Rather than relying on hand-engineered reward functions, this approach leverages human preferences to learn a reward model. Specifically, a human can easily provide preference between a pair of trajectories by the agent, thereby implicitly indicating the desired behaviors or the task's objectives. By learning from human feedback, the agent is capable of accomplishing specific tasks or mastering certain behaviors. Recent research in the field of preference-based RL has demonstrated that it can

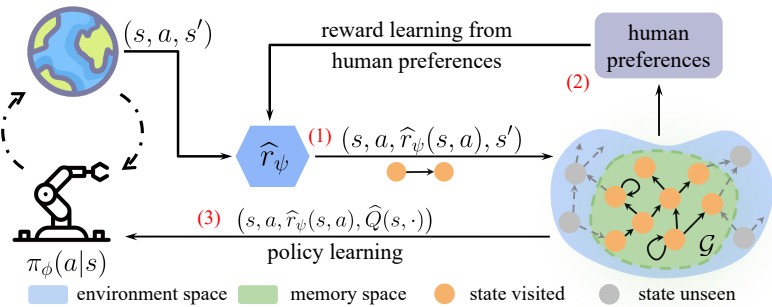

Figure 1: Framework of SEER. (1) Label rewards using $\widehat{r}_\psi$ and construct a non-parametric model $\mathcal{G}$. (2) Query human preferences and update the reward model $\widehat{r}_\psi$ based on them. (3) Update $Q_\theta$ with regularizing $\widehat{Q}$ bootstrapped from $\mathcal{G}$ and optimize policy $\pi_\phi$.

train agents to exhibit novel behaviors and, to some extent, mitigate the challenges of reward hacking. However, existing methods still suffer from feedback inefficiency, limiting the applicability of preference-based RL in practical scenarios and making its deployment more challenging.

PbRL involves sampling informative trajectories, learning a reward model from preferences, optimizing policy with the reward model, and subsequently generating higher-quality trajectories for the next iteration, thereby creating a virtuous circle. Prior research (Hejna & Sadigh, 2023; Liu et al., 2022) observes that the intrinsic inefficiency of reward learning mechanisms results in an increase in feedback requirements for PbRL. Specifically, a limited set of preference labels can lead to an imprecise reward function. This inaccuracy in the reward function may cause the Q function to overfit the erroneous outputs of the reward model, resulting in suboptimal policy, a phenomenon often referred to as confirmation bias (Pham et al., 2021). Further, the data coverage in the replay memory is limited to a tiny subset of the whole state-action space. When combined with deep neural networks, the extrapolation of function approximation may erroneously overestimate out-of-distribution state-action pairs to have unrealistic values, and the errors will be back-propagated to previous states (Fujimoto et al., 2019; Kumar et al., 2019; Levine et al., 2020). In an inaccurate reward model compounded with overestimation bias, a suboptimal policy is extracted from the Q function that cannot reflect the current knowledge in the replay memory, especially at the beginning of learning. The suboptimal policy leads to poor trajectory sampling, low-quality human feedback, less useful reward learning, and thus an inefficient PbRL systm, which indicates the necessity for a more accurate Q function aligned with human preferences, improving this learning cycle.

In this work, we present SEER, an efficient framework via aligned e**S**timation from **ExpE**rience for preference-based **R**einforcement learning. By leveraging historical trajectories, it constructs a non-parametric model for bootstrapping an empirical Q function to enhance and accelerate policy learning. The advantages of the empirical Q function can be categorized into two main areas. Firstly, it aids SEER in selecting informative pairs for efficient querying. Secondly, it regularizes the neural Q function, leading to a policy that is consistent with human preference. Empirical evaluations across a range of puzzle-solving and construction tasks show that SEER significantly outperforms the baselines, enhancing feedback efficiency. This superiority is especially pronounced when available human feedback is limited.

The main contributions of our work can be summarized as follows: First, we propose SEER, a novel feedback-efficient preference-based RL algorithm that utilizes an empirical Q function to assist in sampling and regularizes the neural Q function for efficient learning. Second, experiments demonstrate that our method outperforms other state-of-the-art preference-based RL methods and substantially improves feedback efficiency on various complex tasks. Lastly, leveraging aligned experience estimation, SEER demonstrates an obvious advantage over baselines, especially when limited human feedback is available. We also show that SEER can train an accurate Q function and a better policy.

## 2 PRELIMINARY

**Preference-based Reinforcement Learning.** In the reinforcement learning paradigm, a finite Markov decision process (MDP) is characterized by the tuple $\langle \mathcal{S}, \mathcal{A}, \mathcal{R}, \mathcal{P}, \gamma \rangle$. This includes the

state space $\mathcal{S}$, action space $\mathcal{A}$, transition dynamics, reward function, and discount factor. Specifically, $\mathcal{P}(s'|s, a)$ represents the stochastic dynamics of the environment, indicating the probability of transitioning to $s'$ when action $a$ is taken in state $s$. Meanwhile, $\mathcal{R}(s, a)$ denotes the reward obtained by selecting action $a$ in state $s$. The policy, $\pi(a|s)$, maps the state space to the action space. The objective of the agent is to collect trajectories by interacting with the environment, aiming to maximize the expected return.

In the general framework of preference-based RL from Christiano et al. (2017), there is no reward function from reward engineering; instead, a reward function estimator $\widehat{r}_\psi$ should be learned to be consistent with the preferences of the human expert. Specifically, a segment $\sigma$ is a sequence of states and actions, which is $(s_{t+1}, a_{t+1}, \cdots, s_{t+k}, a_{t+k})$. A human expert provides a preference $y$ on given two segments $(\sigma^0, \sigma^1)$, and $y$ is the distribution over $\{0, 1\}$, $y \in \{(1, 0), (0, 1), (0.5, 0.5)\}$. Following the Bradley-Terry model (Bradley & Terry, 1952), a preference predictor constructed by the estimate of the reward function $\widehat{r}_\psi$ is formulated as follows:

$$P_\psi[\sigma^0 \succ \sigma^1] = \frac{\exp \sum_t \widehat{r}_\psi(s_t^0, a_t^0)}{\exp \sum_t \widehat{r}_\psi(s_t^0, a_t^0) + \exp \sum_t \widehat{r}_\psi(s_t^1, a_t^1)}, \tag{1}$$

where $\sigma^0 \succ \sigma^1$ indicates $\sigma^0$ aligns more closely with human expert expectations than $\sigma^1$. The reward model can be learned by minimizing the cross-entropy loss between predictions from preference predictors and actual human preferences.

$$\mathcal{L}_{\text{reward}}(\psi) = - \mathop{\mathbb{E}}_{(\sigma^0, \sigma^1, y) \sim \mathcal{D}} \Big[ y(0) \log P_\psi[\sigma^0 \succ \sigma^1] + y(1) \log P_\psi[\sigma^1 \succ \sigma^0] \Big]. \tag{2}$$

By optimizing the reward function with respect to this loss, segments that align more closely with human preferences receive a higher cumulative reward.

## 3 METHOD

In this section, we formally present SEER: which is easy to be combined with any existing PbRL algorithm to improve feedback efficiency. In the following, we first describe how to construct the non-parametric model including data structure and update rule, then we propose the objective of our method and training details. The full procedure of our algorithm is summarized in Algorithm 1.

### 3.1 NON-PARAMETRIC MODEL CONSTRUCTION

**Model Structure.** We structure the historical trajectories in the replay memory as a non-parametric model, which is a dynamic and directed graph denoted as $\mathcal{G} = (\mathcal{V}, \mathcal{E})$. Naturally, each vertex in the graph stores a state $s$ and its action value estimation $\widehat{Q}(s)$, leading to the vertex set definition: $\mathcal{V} = \{s|(s, \widehat{Q}(s))\}$. As for each directed edge, it represents a transition from state $s$ to $s'$ via action $a$, and it also stores reward estimation $\widehat{r}_\psi(s, a)$ and transition visit counts $N(s, a, s')$ for model updating. The graph edge set is denoted as $\mathcal{E} = \{s \xrightarrow{a} s'|(a, \widehat{r}_\psi(s, a), N(s, a, s'), \widehat{Q}(s, a)\}$. For query efficiency, both vertices and edges have unique keys generated by a hash function, ensuring a query time complexity of $\mathcal{O}(1)$. Additionally, each vertex $v$ maintains an action set $\partial\mathcal{A}(s)$ which denotes actions taken in state $s$ for in-sample updates.

**Model Updating.** Graph updating mainly includes two parts: the first part is statistical data update, and the second part is reward relabeling. For statistical data updating, upon observing a transition $(s, a, \widehat{r}_\psi(s, a), s')$, we add a new vertex and edge according to the data structure described above, initializing $\widehat{Q}(s) = 0$ and $N(s, a, s') = 1$. If the directed edge already exists, we simply increment the visit count: $N(s, a, s') \leftarrow N(s, a, s') + 1$. Upon meeting the model update criteria, a subset of graph vertices $\partial\mathcal{V} \subseteq \mathcal{V}$ is sampled in reverse order, akin to the methods in Rotinov (2019); Lee et al. (2019), for quickly and efficiently updating. To avoid visiting out-of-sample actions during updating, we constrain the max-operator in the update rule to operate over $\partial\mathcal{A}(s)$ rather than the entire action space. We update $\widehat{Q}$ with value iteration as follows:

$$\widehat{Q}(s) \leftarrow \max_{a \in \partial\mathcal{A}(s)} \left( \widehat{r}_\psi(s, a) + \gamma \sum_{s' \in \mathcal{S}} \widehat{p}(s'|s, a)\widehat{Q}(s') \right), \tag{3}$$

where $\widehat{p}(s'|s,a) = N(s,a,s')/\sum_{s'} N(s,a,s')$ is the empirical dynamics in the graph. Based on the update rule Eq. (3), there is a specific and essential property for our method: it never queries values for unseen actions, thus avoiding overestimation. As for the reward relabel, we estimate the reward on every edge in the graph with reward model $\widehat{r}_\psi$ every time this model is updated. In this way, it maximizes the utilization of historical transitions and mitigates the effects of a non-stationary reward function.

**Sampling Informative Trajectories.** In this work, we design a novel strategy for sampling trajectories based on the graph. We initiate by randomly selecting a subset of vertices from the graph as starting points. From these starting points, we proceed with a forward search along the directed edges connected to the initial vertices. When a vertex has multiple child nodes, we determine the next node based on the one with either the highest or lowest action value estimation, or we choose one randomly. This yields a pair of segments that differ from each other, providing a more informative pair. We continue this procedure until the segment achieves a predetermined length. A further advantage of this approach is that it allows us to stitch new trajectories by utilizing the graph's structure.

## 3.2 POLICY LEARNING

In our method, in addition to the above non-parametric model, we learn another parameterized $Q_\theta$ under the framework of maximum entropy reinforcement learning. and the policy $\pi$ is modeled as a neural network with parameters $\phi$.

For each training iteration, it samples batch data $\{(s_t, a_t, s_{t+1}, \widehat{r}_\psi(s_t, a_t), \widehat{Q}(s_t, a_t))\}$ from $\mathcal{G}$. The function $\widehat{Q}$ is bootstrapped from the graph $\mathcal{G}$ constructed by historical trajectories, which is estimated only by using the state-action pairs in the current replay memory. It serves as a lower bound for neural Q function ($Q_\theta$). The $Q_\theta$ should be constrained by $\widehat{Q}$. To achieve this, we introduce a novel loss to regularize and accelerate policy learning. Drawing inspiration from Haarnoja et al. (2018), we formulate the distribution-constrained loss Eq. (4) to mitigate overestimation and extrapolation errors in the neural Q function as follows:

$$\mathcal{L}_{\mathrm{dc}}(\theta) = \mathbb{E}_{s\sim\mathcal{G}}\Big[D_{\mathrm{KL}}\big(\widehat{\pi}(s)\|\pi_{\mathrm{soft}(\theta)}(s)\big)\Big], \tag{4}$$

where the experience policy is denoted by $\widehat{\pi}(s) = \mathrm{Softmax}(\widehat{Q}(s,\cdot))$ and $\pi_{\mathrm{soft}(\theta)}(s) = \mathrm{Softmax}(Q_\theta(s,\cdot))$. It is important to emphasize that the loss only considers the support set $\partial\mathcal{A}(s)$ for a given state $s$.

For policy learning, it contains two iterations, including soft policy evaluation and soft policy improvement. During soft policy evaluation, we combine the soft Bellman residual with the distribution-constrained loss Eq. (4). The parameters $\theta$ of the Q function are optimized as follows:

$$J_Q(\theta) = \mathbb{E}_{\tau_t\sim\mathcal{G}}\Big[\big(Q_\theta(s_t, a_t) - Q_{\mathrm{target}}\big)^2\Big] + \lambda\mathcal{L}_{\mathrm{dc}}(\theta),$$
$$Q_{\mathrm{target}} = \widehat{r}_\psi(s_t, a_t) + \gamma\pi_\phi(a_t|s_t)^{\mathrm{T}}\big[Q_\theta(s_t, a_t) - \alpha\log\pi_\phi(a_t|s_t)\big], \tag{5}$$

where $\lambda$ is weight for $\mathcal{L}_{\mathrm{dc}}$. The soft Q target is calculated by using the full action distribution. $\tau_t = (s_t, a_t, s_{t+1}, \widehat{r}_\psi(s_t, a_t))$ is the transition at time step $t$, $\alpha$ is a learnable temperature parameter that controls the item of entropy.

After the updating of $Q_\theta$, policy $\pi_\phi$ is updated by minimizing the following loss, which considers using an action probability-weighted objective:

$$J_\pi(\phi) = \mathbb{E}_{s_t\sim\mathcal{G}}\Big[\pi_\phi(a_t|s_t)^{\mathrm{T}}\big(\alpha\log\pi_\phi(a_t|s_t) - Q_\theta(s_t)\big)\Big]. \tag{6}$$

Through alternating soft policy evaluation and soft policy improvement, SEER leads to a well-behaved policy.

## 3.3 THEORETICAL ANALYSIS

For completeness, we provide a theoretical analysis to show the property of the empirical Q function, denoted as $\widehat{Q}$. To learn $\widehat{Q}$, we use Eq. (3) based on the graph. This bootstraps only in-distribution

---

**Algorithm 1** SEER

---

**Input:** preference query frequency $K$, number of human's preference labels per session $M$
1: Initialize parameters of $Q_\theta$, $\pi_\phi$, $\widehat{r}_\psi$, and preference dataset $\mathcal{D} \leftarrow \emptyset$
2: Initialize $\mathcal{G}$ and $\pi_\theta$ with unsupervised exploration
3: **for** each iteration **do**
4:     Take action $a_t \sim \pi_\theta$ and collect $s_{t+1}$
5:     // Query preference and Reward learning
6:     **if** iteration % $K == 0$ **then**
7:         **for** each query step **do**
8:             Sample pair of trajectories $(\sigma^0, \sigma^1)$ and query human for $y$
9:             Store preference data into dataset $\mathcal{D} \leftarrow \mathcal{D} \cup \{(\sigma^0, \sigma^1, y)\}$
10:        **end for**
11:       **for** each gradient step **do**
12:           Sample batch $\{(\sigma^0, \sigma^1, y)_i\}_{i=1}^n$ from $\mathcal{D}$
13:           Optimize Eq. (2) to update $\widehat{r}_\psi$
14:       **end for**
15:       Relabel the whole graph $\mathcal{G}$ using $\widehat{r}_\psi$
16:     **end if**
17:     // Graph $\mathcal{G}$ Construction and Updating
18:     Store transition $(s_t, a_t, \widehat{r}_\psi(s_t, a_t), s_{t+1})$ into Graph $\mathcal{G}$
19:     Update $\widehat{Q}$ on each vertex via Eq. (3)
20:     // Policy Learning
21:     Update $Q_\phi$ and $\pi_\theta$ according to Eq. (5) and Eq. (6), respectively.
22: **end for**
**Output:** policy $\pi_\phi$

---

actions, which derives a conservative action value estimate. Comparing with the $Q$ function bootstrapping from the whole action space, $\widehat{Q}$ mitigates extrapolation error from out-of-distribution data and aligns better with encountering rewards, which reflects human preference. The $\widehat{Q}$ serves as a lower bound for $Q$, and converges to the global optimum as the data coverage increases. The full proofs of Theorem 3.1 are presented in Appendix A.

**Theorem 3.1.** *Let $Q_t$ and $\widehat{Q}_t$ denote the $Q$-values learned following Bellman optimality equation and Eq. (3) at time step $t$ respectively. We have $Q_t$ and $\widehat{Q}_t$ converge to fix points $Q^*$ and $\widehat{Q}^*$, $\lim_{t\to\infty} Q_t = Q^*$, $\lim_{t\to\infty} \widehat{Q}_t = \widehat{Q}^*$. Further, $\widehat{Q}^*(s,a) - Q^*(s,a) \leq 0 \; \forall(s,a) \in \mathcal{S} \times \mathcal{A}$. The equation holds if all state-action pairs are visited.*

## 4 EXPERIMENT

In this section, we conduct experiments to compare performance with three recent state-of-the-art preference-based RL approaches on various puzzle games from Sokoban (Schrader, 2018) and flexible construction tasks from CraftEnv (Zhao et al., 2023). More details about the tasks used in our experiments can be found in the Appendix B. More details on tasks can be found in the appendix. In many cases, it is difficult to clearly distinguish the quality of different trajectories in these tasks, making them appear "equally poor" before achieving any meaningful goals, making preference-based learning more challenging.

### 4.1 SETUP

In our experiments, we select the state-of-the-art preference-based RL algorithm PEBBLE (Lee et al., 2021b) as our backbone algorithm, consistent with previous methods. Since PEBBLE employs the SAC (Haarnoja et al., 2018) algorithm for policy learning, we also compare it to SAC using the ground truth reward directly, which serves as an upper bound for both the baseline and our method. We remark that SEER can be seen as an efficient tool that integrates with any preference-based RL algorithms by substituting the reward learning procedure of its backbone method.

**Baselines.** We choose the reward-based algorithm SAC and three state-of-the-art preference-based RL algorithms for comparison:

- SAC (Haarnoja et al., 2018): Soft Actor-Critic is the backbone RL algorithm of the below methods, and it receives reward signals from the environment. So, reward-based SAC is considered the upper bound of all algorithms in preference-based RL, where reward is not provided.

- PEBBLE (Lee et al., 2021b): The method is a preference-based RL method, which combines unsupervised pre-training with reward learning and relabels all past experience once the reward model is updated.

- SURF (Park et al., 2022): The method applies data augmentation to reward learning, specifically utilizing a large amount of unlabeled data by inferring pseudo-labels, which has significantly improved the efficiency of feedback.

- MRN (Liu et al., 2022): The method is the current state-of-the-art algorithm in preference-based RL, and it is aware of the performance of the Q function in reward learning through bi-level optimization.

- SEER (Ours): The proposed method builds a non-parametric model from past transitions. We can then use this model to create a more accurate empirical Q function, which leads to a better iteration circle for PbRL learning.

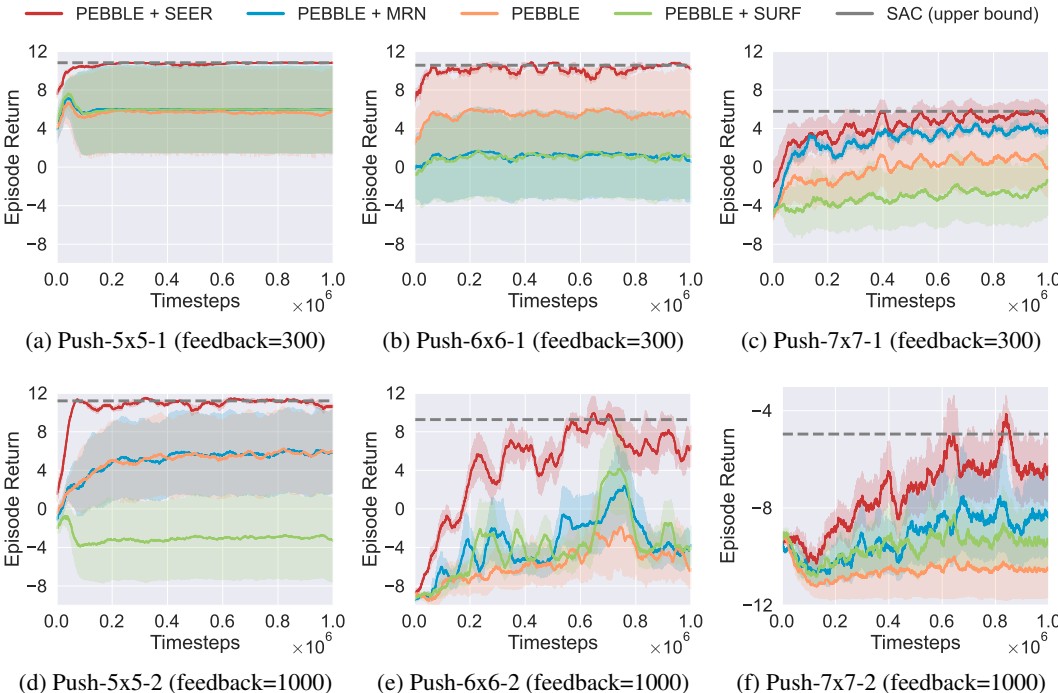

Figure 2: Training curves of various methods on six puzzle solving tasks from Sokoban. The solid line presents the mean values, and the shaded area denotes the standard deviations over five runs.

**Implementation details of SEER.** In the experiments, we use the same basic settings used in Lee et al. (2021b); Park et al. (2022); Liu et al. (2022), including the unsupervised exploration technique and the uncertainty-based trajectories sampling scheme (more details at Appendix C.1). As for the reward learning setting, all methods use an ensemble of three reward models with bounded output to $[-1, 1]$ via a hyperbolic tangent function. Similar to previous research, we consider the same approach for consistent performance evaluation by introducing a scripted teacher. This teacher provides preference labels between two trajectory segments to the agent, based on the inherent reward function. These preferences accurately mirror the environment's ground truth reward, enabling a quantitative assessment of algorithms through the measurement of true returns. However, it's crucial to highlight that the agent cannot directly access this reward in the context of preference-based RL.

We evaluate all methods over five runs independently and report the mean and standard variance of the results for each task. For fair comparisons, all methods train with the same network architecture

and common hyperparameters, except for certain method-specific components. For the number of human preference feedback, we use 300 preference pairs in Push-5×5-1, Push-6x6-1, Push-7×7-1, and 1000 preference pairs for others, including Push-5×5-2, Push-6x6-2, Push-7x7-2, Strip-shaped Building, Block-shaped Building, and Simple Two-Story Building tasks. Within the framework of SEER, we build a non-parametric model. Upon the termination of each episode, it refines this model, specifically updating the $\widehat{Q}$ values on edges and the $\widehat{V}$ values on vertices based on the sampled data.

For the baseline implementation, the approaches PEBBLE [1], SURF [2], and MRN [3] are implemented by using their publicly released code. For each run in our experiments, we deploy a single NVIDIA Tesla V100 GPU and allocate 4 CPU cores for the training process. Further details regarding the implementation of our approach and the aforementioned baselines are elaborated in Appendix C.

## 4.2 RESULTS

**Sokoban experiments.** A detailed introduction and visualization of the six puzzle-solving tasks from Sokoban are presented in Appendix B.1. We selected these tasks for our experiments, covering a range of complexities. Figure 2 depicts the learning curves of the average episode return for SEER and other baselines in Sokoban tasks. For each task, SAC provides the best performance by using the ground-truth reward function as the upper bound of performance. From the learning curves, the performance of SEER outperforms all baselines and presents a significant sample efficiency. Early in the training process, as shown in Figure 2, SEER rapidly achieves outstanding performance across multiple tasks. Interestingly, in several tasks, SEER can approach the performance benchmark with only a few preference labels from humans, showcasing impressive feedback efficiency. We also notice that certain baselines are considerably affected by randomness on certain tasks. This inconsistency results in poor performance in some runs, showing a pronounced variance in results. On the other hand, the training curves of some methods show a decline on more challenging tasks. This indicates that these methods are sensitive to the quality of the collected trajectory pairs. Specifically, if many of the sampled trajectory pairs are of similarly poor quality, it hampers the Q function's learning, culminating in inferior policy performance.

**Craftenv experiments.** For robotic construction tasks, we choose three complex and flexible environments from CraftEnv: Strip-shaped Building, Block-shaped Building, and Simple Two-Story Building task. These tasks require agents to master the manipulation of construction components to realize a desired design, stimulating the complexity of analogous real-world tasks. Task specifics are provided in Appendix B.2. Figure 3 suggests the learning curves of all methods with the same number of human preference labels. As shown in Figure 3, SEER significantly improves the performance of PEBBLE, both in terms of feedback efficiency and algorithmic performance. We also note that SEER achieves comparable performance to PEBBLE using significantly fewer samples. For instance, on the Strip-shaped Building task, SEER exceeds the average performance of PEBBLE using only 30% of the total samples, highlighting an obvious advantage over PEBBLE (represented in orange). These findings suggest that SEER significantly reduces the feedback required for solving complex tasks.

We remark that SEER can effectively leverage state-action pairs in the current replay memory to bootstrap an empirical Q function. This assists SEER in sampling more informative pairs and regularizes the neural Q function, resulting in a policy that is more consistent with human intent. From the comparison of the red (Ours) and orange (PEBBLE) curves in the figures, it is obvious that SEER significantly enhances the performance of PEBBLE. These results from Figure 2 and Figure 3 again demonstrate that SEER improves the feedback-efficiency of preference-based RL methods on a variety of complex tasks.

## 4.3 ABLATION STUDY

**Impact on Preference Amounts.** To evaluate how the quantity of preferences influences the performance of SEER, we carry out an additional experiment to evaluate SEER's efficacy with varying amounts of human preferences. We consider the number of preferences $N \in \{50, 100, 300, 500\}$

---

[1] https://github.com/pokaxpoka/B_Pref
[2] https://github.com/alinlab/SURF
[3] https://github.com/RyanLiu112/MRN

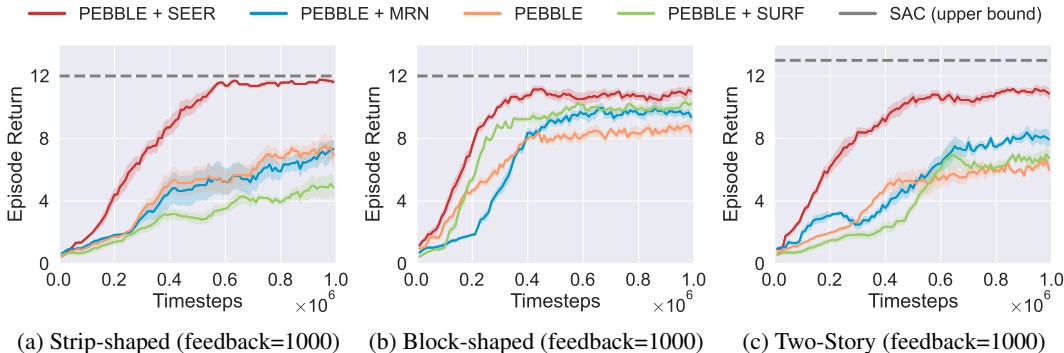

Figure 3: Training curves of all methods on three construction tasks from CraftEnv. The solid line presents the mean values, and the shaded area denotes the standard deviations over five runs.

on Sokoban tasks. The training curves of the average episode return of all methods on tasks are in Figure 4. The results indicate that the performance of the policy gradually improves as the number of preference labels increases. When provided with sufficient preference labels, SEER can approach the performance upper bound. The learning curves depicting the average episode return for all methods across tasks are presented in Figure 4. The results suggest that as the number of preference labels increases, there's a corresponding improvement in the policy's performance. As the provision of preference labels increases, SEER is capable of approaching optimal performance for each task.

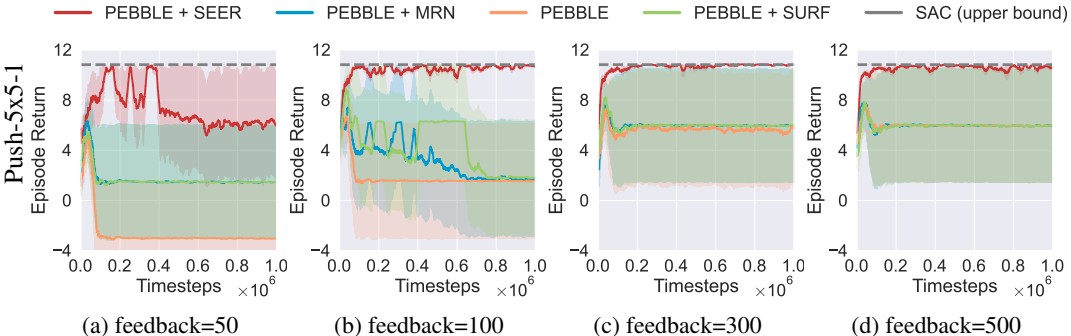

Figure 4: Training curves of all methods with varying numbers of preference labels on Push-5x5-1. The solid line presents the mean values, and the shaded area denotes the standard deviations over five runs.

**Estimation Error of $Q_\theta$.** To further analyze the accuracy of $Q_\theta$, we evaluate the mean squared error (MSE) between ground-truth Q values and $Q_\theta$ trained by various methods. The mean and standard deviation across ten runs are presented in Table 1. SAC serves as the benchmark for $Q_\theta$ quality, and SEER achieves the lowest MSE among all methods. In SEER, $Q_\theta$ is trained with regularization by $\widehat{Q}$, which serves as a lower bound, effectively reducing overestimation and extrapolation errors in $Q_\theta$. Consequently, $Q_\theta$ learned from SEER is notably more accurate than the baselines.

Table 1: MSE of $Q_\theta$ across ten runs.

| Task/Algorithm | PEBBLE | PEBBLE+SURF | PEBBLE+MRN | PEBBLE+SEER (Ours) | SAC |
|---|---|---|---|---|---|
| Push-5x5-1 | $0.40 \pm 0.72$ | $0.39 \pm 0.63$ | $0.32 \pm 0.67$ | $\mathbf{0.06 \pm 0.10}$ | $0.04 \pm 0.03$ |
| Block-shaped | $0.18 \pm 0.05$ | $0.12 \pm 0.04$ | $0.13 \pm 0.04$ | $\mathbf{0.08 \pm 0.03}$ | $0.03 \pm 0.02$ |

## 5 RELATED WORK

**Preference-based Reinforcement Learning.** The idea of learning to solve complex tasks from

human feedback instead of explicit reward has been explored widely. However, directly using human feedback as a reward function or imitating the expert's demonstrations to guide the learning is extremely expensive, given the complexity of some scenarios. Some prior works propose effective approaches that learn a reward model from real humans' preference-based comparisons of agent's behaviors (trajectories) (Christiano et al., 2017; Ibarz et al., 2018). To make things even more efficient, Lee et al. (2021b) suggests a way to learn that combines unsupervised pre-training with reward learning and employs reward relabelling technique. Park et al. (2022) introduces SURF, a semi-supervised reward learning framework that improves reward learning via the pseudo-labels and temporal cropping augmentation. MRN (Liu et al., 2022) incorporates bi-level optimization for improving the quality of Q function. Besides, some researches have various considerations, such as skill extraction (Wang et al., 2022), intrinsic reward (Liang et al., 2022), meta-learning (Hejna III & Sadigh, 2023), policy optimization (Kang et al., 2023), and these methods have improved the efficiency to a certain extent. To evaluate the effectiveness and efficiency, Lee et al. (2021a) presents a benchmark for preference-based RL. Recently, a trend of using numerous existing data and elaborate models to promote the further development of this field seems to be emerging (Kim et al., 2023; Xue et al., 2023; Verma et al., 2023). Preferences-based RL and large-scale language models mutually promote each other, the former helps the fine-tuning process in LLM (Brown et al., 2020; Stiennon et al., 2020; Wu et al., 2021a; Nakano et al., 2021; Ouyang et al., 2022), while using large-scale models as preference models can undoubtedly raise the ability of solving complex control tasks (Arjona-Medina et al., 2019; Early et al., 2022; Gangwani et al., 2020; Ren et al., 2022). Our approach differs in that we build a non-parametric model for informative querying and bootstrapping an empirical Q function to accelerate policy learning.

**Graph-based Reinforcement Learning.** The graph is a powerful and practical tool to characterize and solve problems. Prior works have applied graph structure to reinforcement learning. Eysenbach et al. (2019) and Zhang et al. (2021) consider constructing replay buffer as a weighted directed graph, improving the planning ability of the agent. Shrestha et al. (2021) proposes the Deep Averagers with Costs MDP, compiled from a static experience dataset. Zhang et al. (2023) constructs a graph-based empirical MDP using replay memory, achieving a desirable experimental result by combining it with conservative estimation. Goal-oriented reinforcement learning (GoRL) allows the agent to enhance the ability to tackle long-horizon and sparse reward tasks by generating subgoals. In GoRL, it is a feasible solution to construct abstraction graphs using experience, state-transitions, or observations (Eysenbach et al., 2019; Shang et al., 2019; Huang et al., 2019; Emmons et al., 2020; Zhu* et al., 2020). Jin et al. (2022) builds a dynamical graph based on collected transitions and designs a new sampling method, achieving a more efficient exploration strategy. Furthermore, Lee et al. (2022) presents a method to decouple the connection between high-level control and low-level control with a graph in the setting of multi-level control of goal-conditioned RL. In addition, some works (Zhu et al., 2023; Zhang et al., 2021) focus on learning a graph-based world model, aiming to reduce the difficulty of policy learning.

## 6 CONCLUSION

In this work, we introduce SEER, an innovative preference-based RL framework with enhanced feedback efficiency. We note that SEER is streamlined and lightweight, readily serving as a valuable tool to enhance any preference-based RL approaches. It utilizes historical trajectories to build a non-parametric model, bootstrapping an empirical Q function to constrain the neural Q function, leading to accelerating and improving policy learning. Overall, the benefits provided by the empirical Q function may boil down to two directions. On the one hand, it assists SEER in sampling more informative pairs for effective querying. On the other hand, it serves as lower bounds for the performance of the neural Q function and regularizes it during learning, resulting in a policy more aligned with human intent. The results of experiments on various puzzle-solving tasks and complex construction tasks demonstrate that SEER outperforms the baselines by a large margin and considerably improves the feedback efficiency. The performance gap between our proposed method and the baseline is notably evident, particularly when only a minimal amount of human feedback is accessible. In conclusion, the findings presented in this work offer a fresh perspective on the feedback efficiency improvement of preference-based RL, shedding light on previously unexplored dimensions. We believe that the proposed method can serve as a valuable asset for researchers and practitioners in preference-based reinforcement learning. We hope that our work inspires further investigations, fostering collaborative efforts and innovative approaches in the quest for deeper insights.

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

# A   PROOF OF THEOREM

## A.1   PROOF OF THEOREM 3.1

In this section, we analyze the property of $\widehat{Q}$ in finite state-action space $\mathcal{S} \times \mathcal{A}$. The proof of $\lim_{t \to \infty} Q_t = Q^*$ has been well-established in previous work (Robbins & Monro, 1951; Jaakkola et al., 1993; Melo, 2001). Then the proof of $\lim_{t \to \infty} \widehat{Q}_t = \widehat{Q}^*$ is similar. We first prove the empirical Bellman operator Eq. (9) is a $\gamma$-contraction operator under the supremum norm. Then when updating in a sampling manner as Eq. (10), it can be considered as a random process. Borrowing an auxiliary result from stochastic approximation, we prove it satisfies the conditions that guarantee convergence. Finally, to prove $\widehat{Q}^*$ lower-bounds $Q^*$, we rewrite $\widehat{Q}^*(s,a) - Q^*(s,a)$ based on the standard and empirical Bellman operators. When the data covers the whole state-action space, we naturally have $\widehat{Q}^* = Q^*$.

For proof simplicity, we use $\beta$ denotes policies that interact with the environment and form the current replay memory. We first show existing results for Bellman learning in Eq. (8), and then prove Theorem 3.1 in three steps. The Bellman (optimality) operator $\mathcal{B}$ is defined as:

$$(\mathcal{B}Q)(s,a) = \sum_{s' \in \mathcal{S}} P(s'|s,a)[r + \gamma \max_{a'} Q(s',a')]. \tag{7}$$

Previous works have shown the operator $\mathcal{B}$ is a $\gamma$-contraction with respect to supremum norm:

$$\|\mathcal{B}Q_1 - \mathcal{B}Q_2\|_\infty \le \gamma \|Q_1 - Q_2\|_\infty,$$

the supremum norm $\|v\|_\infty = \max_{1 \le i \le d} |v_i|$, $d$ is the dimension of vector $v$. Following Banach's fixed-point theorem, $Q$ converges to optimal action value $Q^*$ if we consecutively apply operator $\mathcal{B}$ to $Q$, $\lim_{n \to \infty} (\mathcal{B})^n Q = Q^*$.

Further, the update rule in Eq. (8), i.e. $Q$-learning, is a sampling version that applies the $\gamma$-contraction operator $\mathcal{B}$ to $Q$.

$$Q(s,a) \leftarrow r(s,a) + \gamma \max_{a'} Q(s',a'). \tag{8}$$

It can be considered as a random process and will converge to $Q^*$, $\lim_{t \to \infty} Q_t = Q^*$, with some mild conditions (Szepesvári, 2010; Robbins & Monro, 1951; Jaakkola et al., 1993; Melo, 2001).

Similarly, we define the empirical Bellman (optimality) operator $\hat{\mathcal{B}}$ as:

$$(\hat{\mathcal{B}}\widehat{Q})(s,a) = \sum_{s' \in \mathcal{S}} P(s'|s,a)[r + \gamma \max_{a':\beta(a'|s')>0} \widehat{Q}(s',a')]. \tag{9}$$

And the sampling version we used on the graph is:

$$\widehat{Q}(s,a) \leftarrow r + \gamma \max_{a':\beta(a'|s')>0} \widehat{Q}(s',a'), \tag{10}$$

We split Theorem 3.1 into three lemmas. We first show $\hat{\mathcal{B}}$ is a $\gamma$-contraction operator under supremum norm, thus converges to optimal action value $\widehat{Q}^*$, $\lim_{n \to \infty} (\mathcal{B})^n \widehat{Q} = \widehat{Q}^*$. Then we show the sampling-based update rule in Eq. (10) converges to $\widehat{Q}^*$, $\lim_{t \to \infty} \widehat{Q}_t = \widehat{Q}^*$. Finally, we show $\widehat{Q}^*$ lower-bounds $Q^*$, $\widehat{Q}^*(s,a) - Q^*(s,a) \le 0, \forall (s,a) \in \mathcal{S} \times \mathcal{A}$. And when the data covers the whole state-action space, i.e. $\beta(a|s) > 0$ for all state-action pairs, we naturally have $\widehat{Q}^*(s,a) = Q^*(s,a)$.

**Lemma A.1.** *The operator $\hat{\mathcal{B}}$ defined in Eq. (9) is a $\gamma$-contraction operator under supremum norm,*

$$\|\hat{\mathcal{B}}\widehat{Q}_1 - \hat{\mathcal{B}}\widehat{Q}_2\|_\infty \le \gamma \|\widehat{Q}_1 - \widehat{Q}_2\|_\infty.$$

*Proof.* We can rewrite $\|\hat{\mathcal{B}}\widehat{Q}_1 - \hat{\mathcal{B}}\widehat{Q}_2\|_\infty$ as

$$\|\hat{\mathcal{B}}\widehat{Q}_1 - \hat{\mathcal{B}}\widehat{Q}_2\|_\infty$$

$$= \max_{s,a}\Big| \sum_{s'\in\mathcal{S}} P(s'|s,a)[r + \gamma \max_{a'_1:\beta(a'_1|s')>0} \widehat{Q}_1(s',a'_1)] - P(s'|s,a)[r + \gamma \max_{a'_2:\beta(a'_2|s')>0} \widehat{Q}_2(s',a'_2)]\Big|$$

$$= \max_{s,a}\gamma\Big| \sum_{s'\in\mathcal{S}} P(s'|s,a)[\max_{a'_1:\beta(a'_1|s')>0} \widehat{Q}_1(s',a'_1) - \max_{a'_2:\beta(a'_2|s')>0} \widehat{Q}_2(s',a'_2)]\Big|$$

$$\leq \max_{s,a}\gamma \sum_{s'\in\mathcal{S}} P(s'|s,a)\Big| \max_{a'_1:\beta(a'_1|s')>0} \widehat{Q}_1(s',a'_1) - \max_{a'_2:\beta(a'_2|s')>0} \widehat{Q}_2(s',a'_2)\Big|$$

$$\leq \max_{s,a}\gamma \sum_{s'\in\mathcal{S}} P(s'|s,a) \max_{\tilde{a}:\beta(\tilde{a}|s')>0} \Big| \widehat{Q}_1(s',\tilde{a}) - \widehat{Q}_2(s',\tilde{a})\Big|$$

$$\leq \max_{s,a}\gamma \sum_{s'\in\mathcal{S}} P(s'|s,a) \max_{\tilde{s},\tilde{a}:\beta(\tilde{a}|\tilde{s})>0} \Big| \widehat{Q}_1(\tilde{s},\tilde{a}) - \widehat{Q}_2(\tilde{s},\tilde{a})\Big|$$

$$= \max_{s,a}\gamma \sum_{s'\in\mathcal{S}} P(s'|s,a)\|\widehat{Q}_1 - \widehat{Q}_2\|_\infty$$

$$= \gamma\|\widehat{Q}_1 - \widehat{Q}_2\|_\infty,$$

where the last line follows from $\sum_{s'\in\mathcal{S}} P(s'|s,a) = 1$. $\qquad\square$

To show the sampling-based update rule in Eq. (10) converges to $\widehat{Q}^*$, we borrow an auxiliary result from stochastic approximation (Robbins & Monro, 1951; Jaakkola et al., 1993).

**Theorem A.2.** *The random process* $\{\Delta_t\}$ *taking values in* $\mathbb{R}^n$ *and defined as*

$$\Delta_{t+1}(x) = (1 - \alpha_t(x))\Delta_t(x) + \alpha_t(x)F_t(x) \tag{11}$$

*converges to zero w.p.1 under the following assumptions:*

*(1)* $0 \leq \alpha_t \leq 1, \sum_t \alpha_t(x) = \infty$ *and* $\sum_t \alpha_t^2(x) < \infty$;

*(2)* $\|\mathbb{E}[F_t(x)|\mathcal{F}_t]\|_W \leq \gamma\|\Delta_t\|_W$, *with* $\gamma < 1$;

*(3)* $Var[F_t(x)|\mathcal{F}_t] \leq C(1 + \|\Delta_t\|_W^2)$, *for* $C > 0$.

$W$ is a norm. In our proof it is supremum norm.

*Proof.* See Robbins & Monro (1951); Jaakkola et al. (1993). $\qquad\square$

**Lemma A.3.** *Given any initial estimation* $\widehat{Q}_0$, *the following update rule:*

$$\widehat{Q}_{t+1}(s_t, a_t) = \widehat{Q}_t(s_t, a_t) + \alpha_t(x_t, a_t)[r_t + \gamma \max_{a:\beta(a|s_{t+1})>0} \widehat{Q}_t(s_{t+1}, a) - \widehat{Q}_t(s_t, a_t)], \tag{12}$$

*converges w.p.1 to the optimal action-value function* $\widehat{Q}^*$ *if*

$$0 \leq \alpha_t(s,a) \leq 1, \quad \sum_t \alpha_t(s,a) = \infty \quad and \quad \sum_t \alpha_t^2(s,a) < \infty,$$

*for all* $(s,a) \in \mathcal{S} \times \mathcal{A}$.

*Proof.* Based on Theorem A.2, we prove the update rule in Eq. (12) converges.

Rewrite Eq. (12) as

$$\widehat{Q}_{t+1}(s_t, a_t) = (1 - \alpha_t(s_t, a_t))\widehat{Q}_t(s_t, a_t) + \alpha_t(x_t, a_t)[r_t + \gamma \max_{a:\beta(a|s_{t+1})>0} \widehat{Q}_t(s_{t+1}, a)]$$

Subtract $\widehat{Q}^*(s_t, a_t)$ from both sides:

$$\widehat{Q}_{t+1}(s_t, a_t) - \widehat{Q}^*(s_t, a_t)$$
$$= (1 - \alpha_t(s_t, a_t))(\widehat{Q}_t(s_t, a_t) - \widehat{Q}^*(s_t, a_t)) + \alpha_t(x_t, a_t)[r_t + \gamma \max_{a:\beta(a|s_{t+1})>0} \widehat{Q}_t(s_{t+1}, a) - \widehat{Q}^*(s_t, a_t)]$$

Let

$$\Delta_t(s, a) = \widehat{Q}(s, a) - \widehat{Q}^*(s, a) \tag{13}$$

and

$$F_t(s, a) = r + \gamma \max_{a':\beta(a'|s')>0} \widehat{Q}_t(s', a') - \widehat{Q}^*(s, a). \tag{14}$$

We get the same random process shown in Theorem A.2 Eq. (11). Then, proving $\lim_{t \to \infty} \widehat{Q}_t = \widehat{Q}^*$ is the same as proving $\Delta_t(s, a)$ converges to zero with probability 1. We only need to show the assumptions in Theorem A.2 are satisfied under the definitions of Eqs. (13) and (14).

Theorem A.2 (1) is the same as the condition in Lemma A.3. It is easy to achieve, for example, we can choose $\alpha_t(s, a) = 1/t$.

For Theorem A.2 (2), we have

$$\mathbb{E}[F_t(s, a)|\mathcal{F}_t] = \sum_{s' \in \mathcal{S}} P(s'|s, a)[r + \gamma \max_{a':\beta(a'|s')} \widehat{Q}_t(s', a') - \widehat{Q}^*(s, a)]$$
$$= (\hat{\mathcal{B}}\widehat{Q}_t)(s, a) - \widehat{Q}^*(s, a)$$
$$= (\hat{\mathcal{B}}\widehat{Q}_t)(s, a) - (\hat{\mathcal{B}}\widehat{Q}^*)(s, a)$$

Thus,

$$\|\mathbb{E}[F_t(s, a)|\mathcal{F}_t]\|_\infty = \|(\hat{\mathcal{B}}\widehat{Q}_t) - (\hat{\mathcal{B}}\widehat{Q}^*)\|_\infty$$
$$\leq \gamma \|\widehat{Q}_t - \widehat{Q}^*\|_\infty$$
$$= \gamma \|\Delta_t\|_\infty,$$

with $\gamma < 1$.

For Theorem A.2 (3), we have

$$Var[F_t(s)|\mathcal{F}_t] = \mathbb{E}[F_t(s) - \mathbb{E}[F_t(s)|\mathcal{F}_t]|\mathcal{F}_t]^2$$
$$= \mathbb{E}[F_t(s) - ((\hat{\mathcal{B}}\widehat{Q}_t)(s, a) - (\hat{\mathcal{B}}\widehat{Q}^*)(s, a))]^2$$
$$= \mathbb{E}[r + \gamma \max_{a':\beta(a'|s')>0} \widehat{Q}_t(s', a') - \widehat{Q}^*(s, a) - ((\hat{\mathcal{B}}\widehat{Q}_t)(s, a) - (\hat{\mathcal{B}}\widehat{Q}^*)(s, a))]^2$$
$$= \mathbb{E}[r + \gamma \max_{a':\beta(a'|s')>0} \widehat{Q}_t(s', a') - (\hat{\mathcal{B}}\widehat{Q}_t)(s, a)]^2$$
$$= Var[r + \gamma \max_{a':\beta(a'|s')>0} \widehat{Q}_t(s', a')|\mathcal{F}_t]$$

We add and minus a $\widehat{Q}^*$ term to make it close to the RHS in Theorem A.2 (3):

$$Var[r + \gamma \max_{a':\beta(a'|s')>0} \widehat{Q}^*(s', a') + \gamma \max_{a':\beta(a'|s')>0} \widehat{Q}_t(s', a') - \gamma \max_{a':\beta(a'|s')>0} \widehat{Q}^*(s', a')|\mathcal{F}_t]$$

Since $r$ is bounded, thus $r + \gamma \max_{a':\beta(a'|s')>0} \widehat{Q}^*(s', a')$ is bounded. And clearly the second part $\max_{a':\beta(a'|s')>0} \widehat{Q}_t(s', a') - \max_{a':\beta(a'|s')>0} \widehat{Q}^*(s', a')$ can be bounded by $\|\Delta_t\|_\infty$ with some constant. Thus, we have

$$Var[F_t(s)|\mathcal{F}_t] \leq C(1 + \|\Delta_t\|_\infty^2),$$

for some constant $C > 0$ under supremum norm. Thus, by Theorem A.2, $\Delta_t$ converges to zero w.p.1, i.e., $\widehat{Q}_t$ converges to $\widehat{Q}^*$ w.p.1. □

**Lemma A.4.** *The value estimation obtained by Eq. (9) lower-bounds the value estimation obtained by Eq. (7):*

$$\widehat{Q}^*(s, a) - Q^*(s, a) \leq 0 \tag{15}$$

*for all state-action pairs.*

*Proof.* Following the definition of Eqs. (7) and (9), we can rewrite as

$$\max_{s,a}(\widehat{Q}^*(s,a) - Q^*(s,a))$$

$$= \max_{s,a}(\hat{\mathcal{B}}\widehat{Q}^*(s,a) - \mathcal{B}Q^*(s,a))$$

$$= \max_{s,a}(\sum_{s'\in\mathcal{S}} P(s'|s,a)[r + \gamma \max_{\hat{a}':\beta(\hat{a}'|s')>0} \widehat{Q}^*(s',\hat{a}')] - \sum_{s'\in\mathcal{S}} P(s'|s,a)[r + \gamma \max_{a'} Q^*(s',a')])$$

$$= \max_{s,a}\sum_{s'\in\mathcal{S}} P(s'|s,a)\gamma(\max_{\hat{a}':\beta(\hat{a}'|s')>0} \widehat{Q}^*(s',\hat{a}') - \max_{a'} Q^*(s',a'))$$

$$\leq \max_{s,a}\sum_{s'\in\mathcal{S}} P(s'|s,a)\gamma(\max_{\hat{a}'} \widehat{Q}^*(s',\hat{a}') - \max_{a'} Q^*(s',a'))$$

$$\leq \max_{s,a}\sum_{s'\in\mathcal{S}} P(s'|s,a)\gamma \max_{\tilde{a}}(\widehat{Q}^*(s',\tilde{a}) - Q^*(s',\tilde{a}))$$

$$\leq \max_{s,a}\gamma\sum_{s'\in\mathcal{S}} P(s'|s,a)\max_{\tilde{s},\tilde{a}}(\widehat{Q}^*(\tilde{s},\tilde{a}) - Q^*(\tilde{s},\tilde{a}))$$

$$= \gamma\max_{\tilde{s},\tilde{a}}(\widehat{Q}^*(\tilde{s},\tilde{a}) - Q^*(\tilde{s},\tilde{a})) = \gamma\max_{s,a}(\widehat{Q}^*(s,a) - Q^*(s,a))$$

where the last line follows from $\sum_{s'\in\mathcal{S}} P(s'|s,a) = 1$. Then we have

$$\max_{s,a}(\widehat{Q}^*(s,a) - Q^*(s,a)) \leq \gamma\max_{s,a}(\widehat{Q}^*(s,a) - Q^*(s,a))$$
$$\leq \gamma^2\max_{s,a}(\widehat{Q}^*(s,a) - Q^*(s,a))$$
$$\leq \cdots$$
$$\leq \gamma^n\max_{s,a}(\widehat{Q}^*(s,a) - Q^*(s,a))$$

Take limit for both sides and since $0 < \gamma < 1$, we have $\max_{s,a}(\widehat{Q}^*(s,a) - Q^*(s,a)) \leq 0$.

When $\beta(a|s) > 0$ for all state-action pairs, the two contraction operators $\hat{\mathcal{B}}$ and $\mathcal{B}$ are the same. And based on Banach's fixed-point theorem, there is a unique fixed point. Thus $\widehat{Q}^*(s,a) = Q^*(s,a)$ for all state-action pairs., i.e., $\widehat{Q}^*(s,a) - Q^*(s,a) = 0, (s,a) \in \mathcal{S} \times \mathcal{A}$ holds when $\beta(a|s) > 0$ for all state-action pairs. □

Then, we get Theorem 3.1 proved with Lemmas A.1, A.3 and A.4.

## B    ENVIRONMENT SPECIFICATIONS

### B.1    SOKOBAN

Sokoban (Schrader, 2018), the Japanese word for 'a warehouse keeper', is a puzzle video game, which is analogous to the problem of having an agent in a warehouse push some specified boxes from their initial locations to target locations. Target locations have the same number of boxes. The goal of the game is to manipulate the agent to move all boxes to the target locations. Specifically, the game is played on a rectangular grid called a room, and each cell of the room is either a floor or a wall. At each new episode, the environment will be reset, which means the layout of the room is randomly generated, including the floors, the walls, the target locations, the boxes' initial locations, and the location of the agent. We choose four tasks with different complexities from Push-5×5-1 to Push-6×6-2, which is shown in Figure 5. The numbers in the task name denote respectively the size of the grid and the number of boxes.

**State Space.** The state space consists of all possible images displayed on the screen. Each image has the same size as the map, and using the way of dividing each pixel of the image by 255 to normalize into [0,1], we preprocess the image to the inputting state.

**Action Space.** The action space of Sokoban has a total of eight actions, composed of moving and pushing the box in four directions, which are *left*, *right*, *up*, *down*, *push-left*, *push-right*, *push-up*, *push-down* in detail.

**Reward Setting.** The agent gets a punishment with a -0.1 reward after each time step. Successfully pushing a box to the target location, can get a +1 reward, and if all boxes are laid in the right locations, the agent can obtain an extra +10 reward. We set the max episode steps to 120, which means the cumulative reward during one episode ranges from -12 to 10 plus the number of boxes.

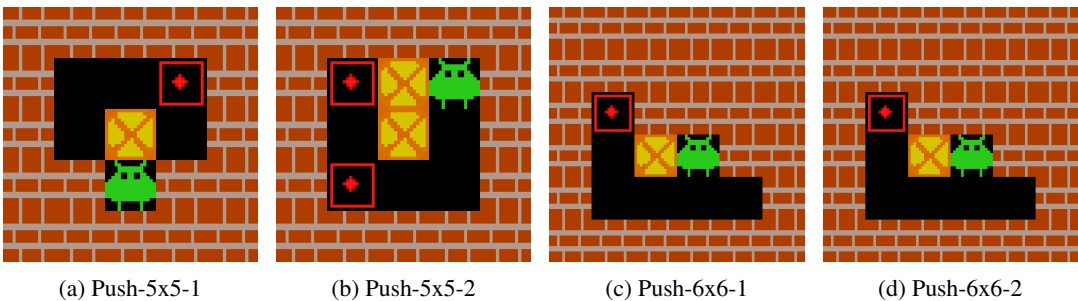

| (a) Push-5x5-1 | (b) Push-5x5-2 | (c) Push-6x6-1 | (d) Push-6x6-2 |

Figure 5: Visualization of puzzle tasks from Sokoban, which focuses on evaluating the capabilities of agents in spatial reasoning, logical deduction, and long-term planning.

### B.2 CRAFTENV

Craftenv (Zhao et al., 2023), A Flexible Robotic Construction Environment, is a collection of construction tasks. The agent needs to learn to manipulate the elements, including smartcar, blocks, and slopes, to achieve a target structure through efficient and effective policy. Each construction task is a simulation of the corresponding complex real-world task, which is challenging enough for reinforcement learning algorithms. Meanwhile, the CraftEnv is highly malleable, enabling researchers to design their own tasks for specific requirements. The environment is simple to use since it is implemented by Python and can be rendered using PyBullet. We choose three different designs of the building tasks, shown in Figure 6, to evaluate our algorithm in CraftEnv.

**State Space.** We assume that the agent can obtain all the information in the map. Therefore, the state consists of all knowledge of smartcar, blocks, folded slopes, unfolded slopes' body, and unfolded slopes' foot, including the position and the yaw angle.

**Action Space.** The available actions of an agent are designed based on real-world smartcar models, including a total of fifteen actions. Besides all eight directions moving actions, i.e. *forward*, *backward*, *left*, *right*, *left-forward*, *left-backward*, *right-forward*, and *right-backward*, there are interaction-related actions, designed to simulate the building process in the real world. Specifically, the agent can act *lift* and *drop* actions to decide whether or not to carry the surrounding basic element, and can *flod* or *unflod* slopes to build the complex buildings. In addition, the actions of *rotate-left* and *rotate-right* control the agent to rotate the main body to the left and right, and *stop* action is just a non-action.

**Reward Setting.** CraftEnv is a flexible environment as mentioned above. We can specify our own reward function for different construction tasks. For the relatively simple tasks of building with specified shape requirement, we can use discrete reward, where some reward is given when part of the blueprint is built. While, for building tasks with high complexity, various reward patterns should be designed to encourage the agent to build with different intentions.

## C EXPERIMENTAL DETAILS

In this section, we provide the implementation details including basic settings for preference-based RL, architecture of neural network, hyper-parameters and other training detail.

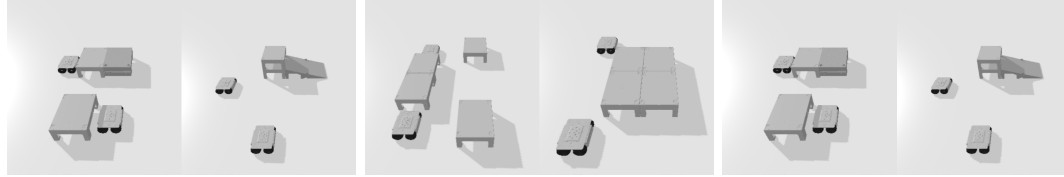

(a) The Strip-shaped Building  (b) The Block-shaped Building  (c) The Simple Two-Story Building

Figure 6: Visualization of building tasks from CraftEnv. From left to right are The Strip-shaped Building, The Block-shaped Building, and The Simple Two-Story Building task respectively.

## C.1 BASIC SETTINGS

In the following section, we provide more details of the unsupervised exploration and the uncertainty-based sampling scheme, both of which are mentioned in Section 4.1. These are pivotal techniques in enhancing the feedback efficiency of algorithms, as referenced in Lee et al. (2021b). To ensure an fair comparison, all preference-based RL algorithms in our experiments incorporate both unsupervised exploration and uncertainty-based sampling.

**Unsupervised Exploration.** The technique of unsupervised exploration in preference-based RL is proposed by Lee et al. (2021b). Designing an intrinsic reward based on the entropy of the state efficiently encourages the agent to visit more diverse states and generate more various behaviors. More specifically, it uses a variant of particle-based entropy (Misra et al., 2003) as the estimation of entropy for the convenience of computation.

**Uncertainty-based Sampling.** There are some different sampling schemes, including but not limited to uniform sampling, disagreement sampling, and entropy sampling. The latter two sampling schemes are classified as uncertainty-based sampling, which has a better performance compared to uniform sampling intuitively and empirically.

## C.2 ARCHITECTURE AND HYPERPARAMETERS.

In this section, we describe the architecture of neural networks of the SAC algorithm, which is used as the underlying model. Then we present the full list of hyperparameters of SAC, PEBBLE, and the proposed SEER. The actor of SAC has three layers, specifically, the first layer is the convolutional layer, composed of 16 kernels with a size of 3. Then we squeeze the output into one dimension as the input for the last two fully connected layers. The two Q networks of SAC have the same architecture as that of the actor, one convolutional layer and two fully connected layers. The detailed parameters of the neural network and hyperparameters during learning are shown in table 2. The hyperparameters of PEBBLE and SEER, which are different from those of SAC, are presented in table 3.

Table 2: Hyperparameters of SAC.

| Hyperparameter | Value | Hyperparameter | Value |
|---|---|---|---|
| Number of layers | 3 layers: 1 Conv2d, 2 Linear | Discount | 0.99 |
| Number of kernels of Conv2d | 16 | Batch size | 256 |
| Size of Kernel of Conv2d | 3 | Initial temperature | 0.2 |
| Stride of Conv2d | 1 | $(\beta_1, \beta_2)$ | (0.9,0.999) |
| Padding of Conv2d | 0 | Update freq | 4 |
| Hidden units of hidden layer | 128 | Critic target update freq | 8000 |
| Activation Function | ReLU | Critic $\tau$ | 1 |
| Actor optimizer | Adam | Exploration | 1 |
| Critic optimizer | Adam | Graph $\tau$ (Graph-based) | 1.0 |
| Learning rate | 1e-4 | Policy weight (Graph-based) | 1.0 |

Table 3: Hyperparameters of PEBBLE and SEER.

| Hyperparameter | Value | Hyperparameter | Value |
|---|---|---|---|
| Length of segment | 50 | Numbers of reward functions / Ensemble size | 3 |
| Learning rate | 0.0003 | Top-k | 5 |
| Reward batch size | 128 | Length of segment (SEER) | 20 |
| Reward update | 200 | Beta $\beta$ (SEER) | 0.5 |
| Frequency of feedback | 2000 | Graph update batch size (SEER) | 32 |
| Number of train steps | 1e6 | Critic update batch size (SEER) | 64 |
| Replay buffer capacity | 1e6 | $\mathcal{L}_{dc}$ weight $\lambda$ | 1 |

