# OpenReview forum: "SEER: Towards Efficient Preference-based Reinforcement Learning via Aligned Experience Estimation"
_ICLR.cc/2024/Conference — Submitted to ICLR 2024_

### Official Review · Reviewer_Atku · 2023-10-27

**Soundness:** 2 fair
**Presentation:** 2 fair
**Contribution:** 2 fair
**Rating:** 3
**Confidence:** 3

**Summary:**

It is known in PbRL that the inaccurate reward model compounding with overestimation bias on Q functions can lead to sub-optimal policies, and subsequently cause poor trajectory sampling and low feedback efficiency. This paper tackles this issue by proposing a non-parametric, graph-based model to learn a lower bound empirical Q-value, and then combined it with a SAC-style policy learning objective. Although the motivation is good, the proposed method suffers from a number of noticeable drawbacks. Please see the following strengths and weaknesses for detailed comments.

**Strengths:**

- The compounding issue of inaccurate reward and overestimation bias is an important problem in PbRL, and worth investigating.
- The idea of correcting overestimated Q values in online PbRL is interesting.
- The proposed method has shown reasonable performance and low variance in the test environments.

**Weaknesses:**

- One of the biggest problems of this paper is that its graph-structured non-parameterized model is only applicable to tasks with discrete states space, or tasks with states and actions that can be enumerated (like the image-based task considered in this paper, there are only finite possible image outcomes). For general continuous control tasks, this method is very impractical. This limits the technical contribution of this work.
- The reason to introduce the non-parametric model as claimed in this paper, is to avoid query unseen actions and avoid overestimation. If this is the case, then why not consider incorporating well-established techniques from offline RL on the replay buffer, like in [1]? Some in-sample learning offline RL methods such as IQL[2] (and a few other methods) can achieve exactly the same purpose but in a much simpler way. The only extra benefit of adopting a graph-based model is the ability to sample new trajectories from the graph, but this part is not carefully ablated, and we do not know whether the performance gains are primarily due to non-overestimated Q values or trajectory sampling.
- Most of the proposed method is to prevent the overestimation of Q values, and it is only weakly relevant to the PbRL problem. Of course, the inaccurate rewards in PbRL can cause the overestimation issue to have a greater impact, but I do not see too many technical designs that are specifically designed for the PbRL problem.
- As for Theorem 3.1, the authors only proved $\hat{Q}$ will lower bound and converge to $Q^*$ learned using Bellman optimality equation under tabular case. However, it says nothing about the property of the final Q-value learned using Eq.(5). There is no analysis on the final Q-value learned with the soft Bellman residual and the distribution-constrained loss, which makes the theoretical analysis insignificant.
- The evaluations are only conducted in two special test environments that are compatible with the graph-structured model. Common PbRL benchmarks like B-Pref are not evaluated. I suppose the proposed method simply cannot run on such continuous control tasks.


**References:**

[1] Ji, T., et al. Seizing Serendipity: Exploiting the Value of Past Success in Off-Policy Actor-Critic. arXiv preprint arXiv:2306.02865.

[2] Kostrikov I, Nair A, Levine S. Offline Reinforcement Learning with Implicit Q-Learning ICLR 2022.

[3] Lee, K. et al. B-Pref: Benchmarking Preference-Based Reinforcement Learning. NeurIPS 2021.

**Questions:**

- I suspect that there should be a trade-off weight hyperparameter on $L_{dc}$ in Eq.(5). As the bellman loss and the regularization term $L_{dc}$ need to be properly balanced to enable stable learning. Have you used a weight hyperparameter here? And if so, how is it tuned?
- The variances in PbRL methods are typically large, but the reported variance of the proposed method is surprisingly small. Why is that?
- The preferences in the experiments are collected from script teachers. How will the method perform if the preference labels come from humans, which contain more noise?

---

### Official Review · Reviewer_pCcb · 2023-10-30

**Soundness:** 2 fair
**Presentation:** 3 good
**Contribution:** 2 fair
**Rating:** 5
**Confidence:** 2

**Summary:**

This paper considers preference-based reinforcement learning and proposes a new policy learning algorithm which can fit into the current framework of PbRL. In each iteration, it first constructs an empirical estimation of the optimal policy and its corresponding distribution-constrained loss. Then it computes the Q function of the current policy via soft Bellman residual and the distribution-constrained loss. After that it applies soft improvement to the current policy with the estimated Q function.

The authors prove that the empirical Q function is an asymptotical lower bound of the optimal Q function. They also conduct numerical experiments to validate the performance of PEBBLE with the proposed policy learning algorithm.

**Strengths:**

The proposed algorithm seems to achieve better performance with PEBBLE in the empirical tasks than SOTA.

**Weaknesses:**

(1) The proposed algorithm seems to only modify the existing framework of maximum entropy RL a little bit and not very novel.

(2) The paper only shows that empirical Q function is a lower bound of the ground-truth optimal Q function. I think the distribution-constrained loss is more reasonable if the authors can further show the empirical Q function is close to the ground-truth optimal Q function.

**Questions:**

Why do the authors use the KL divergence between $\hat{\pi}(s)$  and $\pi_{ soft(\theta)}(s)$ as the regularization term? A more intuitive choice would be the direct distance between $Q_{\theta}$ and $\hat{Q}$.

---

> ### Author Response · Authors · 2023-11-17
> **Response to Reviewer pCcb**
>
> We thank reviewer pCcb for the constructive comments. We value your insights and address each point you raised. Below are our point-wise responses:
>
> **Q1. The proposed algorithm seems to only modify the existing framework of maximum entropy RL a little bit and not very novel.**
> > **A1**: Thanks for your question. The general framework of PbRL consists of the reward learning and policy learning. SEER employs SAC as its backbone algorithm in policy learning, similar to baselines like PEBBLE [1], SURF [2], and MRN [3]. Main Contirbutions lie in, SEER leverages the non-parametric model to sample informative trajectories and regularize neural Q function for feedback-efficient learning.
>
> **Q2. The paper only shows that empirical Q function is a lower bound of the ground-truth optimal Q function. I think the distribution-constrained loss is more reasonable if the authors can further show the empirical Q function is close to the ground-truth optimal Q function.**
> > **A2**: Thank you for your question. In response to your concern about the empirical Q function's closeness to the ground-truth optimal Q function, we have conducted additional experiments. The results are as follows:
> >
> Table 1: MSE of $\hat{Q}$ and $Q_\theta$ across ten runs
> | Task/Q | $\hat{Q}$ | $Q_\theta$|
> | -------- | -------- | -------- |
> | Push-5x5-1   | 0.05±0.07  | 0.06±0.10 |
> | Block-shaped | 0.07±0.03  | 0.08 ±0.03|
>
> >This result represents the performance of the empirical Q and the neural Q  across different tasks. As indicated by the result, the empirical Q closely approximates the ground-truth optimal Q value.
>
> **Q3. Why do the authors use the KL divergence between $\hat\pi(s)$ and $\pi_{soft(\theta)}(s)$ as the regularization term? A more intuitive choice would be the direct distance between $Q_{\theta}$ and $\hat{Q}$.**
> > **A3**: Thanks for your thoughtful question. The decision to use KL divergence is rooted in our objective to align the learned policy as closely as possible with the optimal policy derived from the Q-function. This choice is inspired by the SAC, where the KL divergence is used to minimize the distance between the current policy and the exponentiated softmax of the Q-function, as detailed in [4] Equation 4.
>
> **Reference**: \
> [1] PEBBLE: Feedback-efficient interactive reinforcement learning via relabeling experience and unsupervised pre-training. ICML, 2021. \
> [2] SURF: Semi-supervised reward learning with data augmentation for feedback-efficient preference-based reinforcement learning. ICLR, 2022. \
> [3] Meta-Reward-Net: Implicitly Differentiable Reward Learning for Preference-based Reinforcement Learning. NeurIPS, 2022. \
> [4] Soft Actor-Critic Algorithms and Applications. ICML, 2018.

---

> > ### Comment · Reviewer_pCcb · 2023-11-21
> >
> > Thanks for the response! I still feel the novelty of this work is limited, so I will maintain my score.

---

> > > ### Author Response · Authors · 2023-11-22
> > > **Response to Reviewer pCcb**
> > >
> > > We thank the reviewer for comments on the novelty of our work. We believe the novelty of our approach to alleviating confirmation bias through a non-parametric model is underscored by our findings that regularizers based on conservative estimation can significantly enhance feedback efficiency in PbRL. This contribution is particularly important given the central role of the Q function in the learning cycle of PbRL. While previous work [1] has indeed emphasized the importance of the quality of the Q function, SEER introduces a novel direction by focusing on improving the use of replay memory and empirical estimation regularization. We are open to expanding on these points in our manuscript to better highlight our novel contributions.
> > >
> > > **Reference**: \
> > > [1] Meta-Reward-Net: Implicitly Differentiable Reward Learning for Preference-based Reinforcement Learning. NeurIPS, 2022.

---

### Official Review · Reviewer_pgTP · 2023-11-01

**Soundness:** 3 good
**Presentation:** 3 good
**Contribution:** 3 good
**Rating:** 5
**Confidence:** 3

**Summary:**

The paper introduces SEER, an efficient framework for preference-based Reinforcement Learning (PbRL). Traditional RL poses challenges in creating reward functions. SEER tackles this problem by learning rewards based on human preferences among various trajectories, creating a virtuous circle of learning. The main innovation lies in the empirical Q function derived from past trajectories, improving the sampling process and policy regularization. Experimental results demonstrate SEER's significant edge over state-of-the-art methods, particularly in scenarios with limited human feedback.

**Strengths:**

1. Innovative Approach: SEER presents a novel method of leveraging historical trajectories to construct an empirical Q function. This Q function aids in bootstrapping and enhances policy learning.
2. Efficiency: On the domains tested, SEER appears to outperform baselines.
3. Theoretical Underpinning: The paper provides a theoretical demonstration that the empirical Q function serves as a lower-bound of the oracle Q under human preference.

**Weaknesses:**

The main weakness with the paper is in the experimental evaluation. All of the other baselines test on the same broad suite of simulated robotics tasks. Since this paper follows from those papers and directly compares against them, those environments should definitely be included in the evaluation.

Another weakness is with the presentation. There are a lot of grammar mistakes throughout, for example the first sentence in abstract should be “One of the challenges… “ also in abstract “optimizing policies”, also citation (III and Sadigh) should be Hejna III and Sadigh. I think related works could be improved a lot. For example, the main baseline methods are not well-described in this section.

**Questions:**

Evaluation: How well does SEER do on the benchmarks that are common in the literature? Is there a reason those benchmarks are not included?
Generalizability: How adaptable is SEER across different domains or problems? Can it be seamlessly integrated into other existing RL algorithms?
Human Feedback: How does SEER handle potentially conflicting or inconsistent human preferences? Is there a mechanism to resolve such conflicts?

---

> ### Author Response · Authors · 2023-11-17
> **Response to Reviewer pgTP**
>
> We thank reviewer pgTP for the constructive comments. We will give our point-wise responses below.
>
> **Q1. The main weakness with the paper is in the experimental evaluation. All of the other baselines test on the same broad suite of simulated robotics tasks. Since this paper follows from those papers and directly compares against them, those environments should definitely be included in the evaluation.**
> > **A1**: Thank you for your valuable feedback. SEER primarily utilizes a discrete graph for conservative estimations, which inherently aligns more with discrete action domains and therefore is not very suitable for continuous action domains. However, we highlight that SEER demonstrates the benefit of regularization based on conservative estimation for feedback-efficient learning. This finding opens up a wide range of research possibilities for future studies. In terms of addressing continuous action scenarios, we are considering the adoption of alternative approaches, such as offline RL methods [1], to achieve conservative estimation without explicitly constructing a graph.
>
>
> **Q2. Another weakness is with the presentation. There are a lot of grammar mistakes throughout, for example the first sentence in abstract should be “One of the challenges… “ also in abstract “optimizing policies”, also citation (III and Sadigh) should be Hejna III and Sadigh. I think related works could be improved a lot. For example, the main baseline methods are not well-described in this section.**
> > **A2**: Thank you for your valuable feedback regarding the presentation aspects of our paper. We will review the entire manuscript to correct these issues. Regarding the citations, we will ensure that all references, such as 'Hejna III and Sadigh', are correctly formatted and cited according to the required style guidelines.
>
>
> **Q4. How adaptable is SEER across different domains or problems? Can it be seamlessly integrated into other existing RL algorithms?**
> > **A4**: Thanks for your question. SEER uses a discrete graph to obtain conservative estimations, which is not very suitable for continuous action domains. However, we have strategies to extend its application to continuous action scenarios. Firstly, one approach is to employ discretization techniques to transform continuous action spaces into discrete equivalents. This method is commonly used in contexts such as robotic arm manipulation, exemplified by RT-2 [2], and in complex games like DOTA [3] and StarCraft [4]. Secondly, SEER can be adapted using an action translator for generating continuous actions, such as the VMG model [5], involves using a graph-structured world model to convert actions via an action translator. The PbRL framework generally contains reward learning and policy learning, with the latter often directly employing standard RL algorithms. SEER is designed to be compatible with any value-based RL algorithm during the policy learning stage.
>
>
> **Q5. How does SEER handle potentially conflicting or inconsistent human preferences? Is there a mechanism to resolve such conflicts?**
> > **A5**: Thanks for pointing our this. While directly resolving such conflicts was not the primary focus of this work, we recognize its importance in the robustness of SEER. To address this, we have conducted experiments to understand how SEER performs under conditions of noisy or conflicting data, which can be seen as a proxy for inconsistent human preferences. Specifically, we tested SEER in Push-5x5-1 and Strip-shaped with varying ratios of noisy data, including {0%, 10%, 15%, 20%, 25%, 30%}. This noisy data simulates scenarios where human preferences might be inconsistent or conflicting. The results, as shown in the following tables, indicate that SEER's performance degrades with increasing levels of noise. It fails with 30% noisy labels in Push-5x5-1 and with 25% noisy data in Strip-shaped.
>
> Table 1: Performance of SEER with different noise levels
> | Noise lable|0\%   | 5\%  |10\%  | 15\% |20\%  |25\%  | 30\% |
> | --------   | ---- | ---- | ---- | ---- | ---- | ---- | ---- |
> | Push-5x5-1 | 10.8 | 10.6 | 9.3  | 9.3  | 8.7  | 5.6  | -1.9 |
> |Strip-shaped| 11.4 | 11.2 | 10.5 | 8.0  | 6.6  | 2.1  |   /  |
>
> Table 2: Performance of PEBBLE with different noise levels
> | Noise lable|0\%   | 5\%  |10\%  | 15\% |20\%  |25\%  | 30\% |
> | -   | -|- | - | - | - | - | ---- |
> | Push-5x5-1 | 6.1  | 5.7  | 1.3  |-0.9  | -9.1 | /    |  /   |
> |Strip-shaped| 7.2  | 5.8  | 3.9  | 1.2  | /    | /    |  /   |
>
> **Reference** \
> [1] Offline reinforcement learning: Tutorial, review, and perspectives on open problems. arXiv, 2020. \
> [2] Rt-2: Vision-language-action models transfer web knowledge to robotic control. arXiv, 2023. \
> [3] Dota 2 with large scale deep reinforcement learning. arXiv, 2019. \
> [4] Grandmaster level in StarCraft II using multi-agent reinforcement learning. Nature 575.7782 (2019): 350-354. \
> [5] Value Memory Graph: A Graph-Structured World Model for Offline Reinforcement Learning. ICLR, 2023.

---

> ### Author Response · Authors · 2023-11-21
> **A mild reminder**
>
> We are grateful to you for the feedback. We have endeavored to address the issues highlighted in your review. It would be helpful to know if the revisions we have made align with your expectations and satisfactorily address your concerns. If you find that they do, we would appreciate it if you could consider reflecting this in an updated evaluation of our work. Our aim is to ensure that the changes we have implemented effectively improve the paper's quality. We are fully open to further discussion and eagerly await any additional guidance or questions you may have.

---

> > ### Comment · Reviewer_pgTP · 2023-11-21
> > **Reviewer Response**
> >
> > Thanks for the detailed response. I have bumped my score up but still vote for reject because there isn't experiments on the standard benchmarks.

---

### Official Review · Reviewer_AZLG · 2023-11-02

**Soundness:** 2 fair
**Presentation:** 2 fair
**Contribution:** 3 good
**Rating:** 5
**Confidence:** 2

**Summary:**

The authors introduce a novel framework for preference-based reinforcement learning, asserting that their results enhance label efficiency. They provide experimental verification of this claim.

In my understanding, the proposed framework is roughly as follows.

After unsupervised exploration without rewards, iterate
* 1) Sample from trajectories based on the constructed graph and get the preference feedback
* 2) Update rewards from a pair of trajectories and preferences.
* 3) Update a graph and an empirical Q-function with updated rewards  (in a conservative way)
* 4) Learn a soft Q-function and an associated policy by using soft Bellman loss + regularization based on a policy corresponding to an optimal policy from the empirical conservative Q-function in Step 3.

**Strengths:**

The framework appears to be novel. Experimental results are solid.

**Weaknesses:**

Certain aspects of the paper remain unclear. My primary concern revolves around the justification for the effectiveness of the proposed framework. While there are several intuitive statements provided and there are solid experiments, they often lack the formal exposition for readers to gain a comprehensive understanding.

* I comprehend the author's assertion regarding the conservatism of the empirical Q-function. However, I am seeking clarification regarding the formal properties of the resulting policy, denoted as SAC $\pi_{\phi}$. Are we anticipating it to exhibit conservatism or optimism? Additionally, the author contends that it "aligns with human preference." Could this alignment be elucidated in a more rigorous manner?

* The author state that "theoretically, we demonstrate that the empirical Q-function is a lower-bound of ...." in the Abstract. However, it is challenging to discern the precise details from Theorem 3.1. in a main text. Several elements remain undefined, such as the exact meanings of $Q_t$ and $\hat Q_t$ in the main text, as well as the underlying assumptions (e.g., do we need assumptions for rewards to say $\hat Q_t$ converges to $Q^{\star}$?  )

* The proposed framework appears to be tailored for tabular settings, primarily due to its reliance on an empirical Q-function. How does the author intend to extend this approach to accommodate continuous settings, particularly in terms of both algorithmic and theoretical considerations?

* In a related context, update (3) seems somewhat naive in addressing the data coverage concern. I agree it might be beneficial to differentiate between actions that have not been visited, actions that have been visited. But how can we distinguish actions that fall in between – perhaps those visited frequently versus those visited infrequently?

**Questions:**

I raised several questions in the weakness part. Furthermore,

* Would you explain Line 8 in Algorithm 1 more? How did you sample a pair of trajectories? Does this correspond to the "Sampling informative trajectories part"?

---

> ### Author Response · Authors · 2023-11-17
> **Response to Reviewer AZLG**
>
> Thank you, Reviewer AZLG, for your constructive comments. We appreciate the insights and suggestions provided. Below are our point-wise responses addressing each of your concerns:
>
> **Q1. I comprehend the author's assertion regarding the conservatism of the empirical Q-function. However, I am seeking clarification regarding the formal properties of the resulting policy, denoted as SAC $\pi_\phi$. Are we anticipating it to exhibit conservatism or optimism? Additionally, the author contends that it "aligns with human preference." Could this alignment be elucidated in a more rigorous manner?**
> > **A1**: Thanks for your question. In our approach, the primary objective is to obtain an accurate Q function to enhance policy learning in PbRL. To this end, we employ an empirical Q-value, which embodies a conservative estimation approach, to constrain the neural Q function. This strategy is designed to accelerate and refine the policy learning process. Additionally, the term 'aligns with human preference' in this context implies that a more precise Q-value estimation correlates with greater alignment to human preferences within the PbRL framework.
>
> **Q2. The author state that "theoretically, we demonstrate that the empirical Q-function is a lower-bound of ...." in the Abstract. However, it is challenging to discern the precise details from Theorem 3.1. in a main text. Several elements remain undefined, such as the exact meanings of $Q_t$ and $\hat{Q}_t$ in the main text, as well as the underlying assumptions (e.g., do we need assumptions for rewards to say converges to?)**
> > **A2**: Thank you for your valuable feedback. $Q_t$ represents the Q-values at time step t learned following the Bellman optimality equation, which is a standard approach in reinforcement learning for estimating the expected utility of actions in a given state. $\hat{Q}_t$ denotes the Q-values estimated at the same time step t but derived from our empirical approach as outlined in Eq. (3). The theorem and its proof are constructed under the general RL framework, without additional assumptions about their nature or convergence properties.
>
> **Q3. The proposed framework appears to be tailored for tabular settings, primarily due to its reliance on an empirical Q-function. How does the author intend to extend this approach to accommodate continuous settings, particularly in terms of both algorithmic and theoretical considerations?**
> > **A3**: Thank you for your insightful question. To adapt our framework to continuous settings, there are several strategies. Firstly, it can consider the incorporation of discretization techniques to transform continuous action spaces into discrete equivalents. This method is common in similar contexts; for instance, RT-2 [1] successfully applies it for discretizing actions in robotic arm applications. Secondly, it can also introduce an action translator to generate continuous actions, akin to the VMG model [2], which uses a graph-structured world model and convert generated actions via an action translator.
>
> **Q4. In a related context, update (3) seems somewhat naive in addressing the data coverage concern. I agree it might be beneficial to differentiate between actions that have not been visited, actions that have been visited. But how can we distinguish actions that fall in between – perhaps those visited frequently versus those visited infrequently?**
> > **A4**: Thank you for your valuable feedback regarding update rule (3) in our paper. In our method, we use $N(s,a,s')$ to represent the number of visits to different actions. The empirical dynamics $\widehat{p}(s'|s,a) = N(s, a, s')/\sum_{s'} N(s, a, s')$ already encapsulate a distinction in visit frequencies. For instance, actions that have been visited frequently will have a higher value of $N(s, a, s')$, reflecting in a higher empirical probability, whereas infrequently visited actions will have lower values. This differentiation is crucial for ensuring that our update mechanism does not overlook less explored actions.
>
> **Q5. Would you explain Line 8 in Algorithm 1 more? How did you sample a pair of trajectories? Does this correspond to the "Sampling informative trajectories part"?**
> > **A5**: Thanks for your question. Yes, Line 8 indeed corresponds to the 'Sampling informative trajectories' part of our algorithm.
>
>
> **Reference** \
> [1] Rt-2: Vision-language-action models transfer web knowledge to robotic control. arXiv, 2023. \
> [2] Value Memory Graph: A Graph-Structured World Model for Offline Reinforcement Learning. ICLR, 2023.

---

> > ### Comment · Reviewer_AZLG · 2023-11-21
> >
> > Thank you very much for the detailed response. I will take this into account in the discussion phase. But, I think my big questions are still remaining.
> >
> > * Q1. Answer to "I am seeking clarification regarding the formal properties of the resulting policy.... "
> >
> > I am not sure my intention is properly conveyed. I want to see more quantitative formal explanations, not informal explanations. For example, why can you claim in your response, "This strategy can accelerate and refine the policy learning process. "?  There is a gap. Fundamentally,  my biggest question is the formal properties of resulting output policies. It is still not explained properly.
> >
> > * Q1. Answer to --- Additionally, the author contends that it "aligns with human preference." Could this alignment be elucidated in a more rigorous manner?"-----
> >
> > I feel the author's response is still hard to decode. I am curious how the author really showed "a more precise Q-value estimation correlates with greater alignment to human preferences within the PbRL framework"?   I want to see some theorem here in the ideal case. Or is it just an informal conjecture?
> >
> > * Q3. The author says discretization works in a continuous setting. But, why do we need to bypass empirical Q-functions to get conservative values? Why don't we get conservative values directly? There are tons of more formal works to do that in offline RL without discretization.
> >
> > * Answer to Q2.  Here, my suggestions is to change the way of writing the theorem to a more informal one. "Standard approach " does not mean readers can see what is $\hat Q_t$ exactly. Of course, readers can roughly guess. But, when people write a theorem, every part must be clearly defined.  This would not be a proper way to write a theorem.
> >
> > * Answer to Q2:  Answer to "The theorem and its proof are constructed under the general RL framework, without additional assumptions about their nature or convergence properties."
> >
> >  I am not sure what you mean by "general".  What do you mean by "without additional assumptions "?  I am suggesting you clearly specify in the main text. For example, did you assume environments are tabular?

---

> > > ### Author Response · Authors · 2023-11-22
> > > **Response to Reviewer AZLG**
> > >
> > > - In response to the 'aligning with human preference': In the context of PbRL, the optimal Q is based on the underlying reward behind human intent, and the loss (Eq. (1) in our paper) of this underlying reward will achieve 0. A more precise Q estimation is indicative of a smaller deviation from this optimal Q. Consequently, a more precise Q-value estimation implies a greater alignment with human preferences. This is the logic that underpins our use of the phrase 'a more precise Q-value estimation means greater alignment to human preferences' in our paper.
> > > - For Q3, I understand you're inquiring about the reason behind building a graph to estimate empirical Q-values. Using graphs is a trade-off between accuracy and generativity. Graphs can provide more accurate estimations but tend to have weaker generalization capabilities. On the other hand, employing neural networks to approximate empirical Q networks might result in lower accuracy but offer better generalizability. Furthermore, the computational burden to update the graph is considerably lighter compared to training new neural networks. Specifically, the graph in SEER is updated only at the end of each episode, and the required batch size for these updates is just 32. This batch size is substantially smaller than what is generally used for updating neural networks. Additionally, each update in the graph involves a single node's values, which is a far less computationally intensive process than updating all parameters in a neural network.
> > > - Regarding Theorem 3.1, we acknowledge that it assumes environments are tabular. In light of this, we will revise the theorem section to more clearly state that our analysis is conducted within the context of a finite state-action space. Meanwhile, this revision will include providing the Bellman optimality equation for $Q_t$ to ensure it is clearly defined and understood.

---

> ### Author Response · Authors · 2023-11-21
> **A mild reminder**
>
> We sincerely thank you for your constructive comments. And we would like to know whether we have addressed your concerns. If so, might you be able to update your rating to reveal this?  Our goal is to ensure that our responses and modifications meet your expectations and enhance the quality of our work. We are eager to continue this dialogue and are available for any further discussion that may be helpful.

---

### Author Response · Authors · 2023-11-21
**Common Response**

We sincerely thank all reviewers for their thorough evaluation and constructive feedback on our paper. We are grateful for the recognition of novelty [AZLG, pgTP], efficiency [pgTP, pCcb], and the emphasis on avoiding overestimation bias [Atku] in our work. In response to the valuable suggestions, we have added several new experiments and made revisions to our paper, which are highlighted in blue, to directly address the comments from the reviewers:

1. **Response to Reviewer AZLG**: We have updated Section 3.1 to provide clearer details about the graph model update process.

2. **Response to Reviewer pgTP**:
   - We have corrected grammatical mistakes and ensured proper formatting and citation according to the required style guidelines.
   - The section on related works has been revised to provide a more detailed introduction to the baseline and to highlight the differences between SEER and these baselines.

3. **Response to Reviewer pCcb**: To address concerns about the empirical Q function, we have conducted additional experiments to evaluate its performance and accuracy.

4. **Responses to Reviewers pgTP and Atku**: In response to concerns about conflicting or inconsistent human preferences, we have conducted supplementary experiments. These experiments investigate how SEER performs in scenarios characterized by noisy or conflicting data, adding robustness to our findings.

5. **Responses to Reviewers Atku**: We have updated Equation (5) in our paper to explicitly include the loss weight $\lambda$ and revised Table 3 in Appendix C.2.

---

### Meta-Review · Area_Chair_QZNX · 2023-12-06

**Metareview:**

The authors present SEER, an efficient framework for preference-based reinforcement learning. Traditional RL poses challenges in creating reward functions. SEER tackles this problem by learning rewards based on human preferences among various trajectories, creating a virtuous circle of learning. An empirical Q function is derived from past trajectories, improving the sampling process and policy regularization. The authors prove that the empirical Q function is an asymptotical lower bound of the optimal Q function. They also conduct numerical experiments to validate the performance of PEBBLE with the proposed algorithm.

The reviewers appreciate the superior empirical performance of the frameowrk. There are several concerns, such as limited novelty, unclear justification of the empirical Q-function, limited to discrete space, and insufficient experiments. The rebuttal and new experiments answered some questions but did not fully address the concerns.

**Justification For Why Not Higher Score:**

All four reviewers recommend rejection. There are several concerns that are not fully addressed after rebuttal.

**Justification For Why Not Lower Score:**

N/A

---

### Decision · Program_Chairs · 2024-01-16

Reject